# RETHINKING KNOWLEDGE DISTILLATION: A MIXTURE-OF-EXPERTS PERSPECTIVE

## ABSTRACT

Knowledge distillation (KD) aims to transfer useful information from a large-scale model (teacher) to a lightweight model (student). Classical KD focuses on leveraging the teacher's predictions as soft labels to regularize student training. However, the exact match of predictions in Kullback-Leibler (KL) divergence could be somewhat in conflict with the classification objective, given that the distribution discrepancies between teacher-generated predictions and ground-truth annotations tend to be fairly severe. In this paper, we rethink the role of teacher predictions from a Mixture-of-Experts (MoE) perspective and transfer knowledge by introducing teacher predictions as latent variables to reformulate the classification objective. This MoE strategy results in breaking down the vanilla classification task into a mixture of easier subtasks with the teacher classifier as a gating function to weigh the importance of subtasks. Each subtask is efficiently conquered by distinct experts that are effectively implemented by resorting to multi-level teacher outputs. We further develop a theoretical framework to formulate our method, termed MoE-KD, as an Expectation-Maximization (EM) algorithm and provide proof of the convergence. Extensive experiments manifest that MoE-KD outperforms advanced knowledge distillers on mainstream benchmarks.

## 1 INTRODUCTION

Deep learning has shown its significance by boosting the performance of various real-world tasks such as computer vision (Krizhevsky et al., 2012), natural language processing (Devlin et al., 2018), and reinforcement learning (Silver et al., 2016). However, it is worth mentioning that the effectiveness of deep learning generally comes at the expense of huge computational complexity and massive storage requirements. This greatly restricts the deployment of large-scale models (teachers) in real-time applications where lightweight models (students) are preferable due to limited resources. Under this context, with the primary goal of improving the student's performance for the task at hand, knowledge distillation (KD) (Gou et al., 2021; Wang & Yoon, 2021) is introduced as a de facto standard to transfer knowledge from a teacher model to a student model.

The rationale behind KD can be explained from an optimization perspective: there is evidence that high-capacity models can find good local minima due to over-parameterisation (Du & Lee, 2018; Soltanolkotabi et al., 2018). This motivates KD to use such models to facilitate the optimization of lower-capacity models (i.e., the student) during training. Classically, KD is approached by minimizing the Kullback-Leibler (KL) divergence between predictive distributions of the teacher and student (Hao et al., 2024; Hinton et al., 2015), the motivation behind which is to leverage the teacher's predictions as soft labels to regularize the student training (Müller et al., 2019; Yuan et al., 2020). However, the efficacy of classical KD is challenged by counter-intuitive observations (Cho & Hariharan, 2019; Stanton et al., 2021). Specifically, a larger teacher does not necessarily increase a student's accuracy compared to a relatively smaller teacher. This can be attributed to the capacity gap between the two models which makes the discrepancy between their predictions significantly large (Huang et al., 2022a). On the one hand, some methods (Dong et al., 2023; Mirzadeh et al., 2020; Park et al., 2021; Son et al., 2021) develop student-friendly teachers to tackle the poor learning issue of the student model. Unfortunately, such methods suffer from complex distillation procedures and heavy computational costs for re-training the teacher model, therefore not being applicable in practice. On the other hand, ATS (Li et al., 2022b) separately applies a higher/lower temperature to the correct/wrong class by finding that more complex teachers are more likely to assign a larger

score for the correct class or less varied scores for the wrong classes while KD-Zero (Li et al., 2024) develops automated searches for distillers without manual architecture modification and KD design.

Despite remarkable progress, people tend to overlook the fact that there could be a significantly large discrepancy between ground-truth labels and teacher-generated labels. In particular, whether the temperature is high or low, the teacher would produce imbalanced predictive distributions even though it is trained on a balanced dataset (Niu et al., 2022). Given the fact that classical KD typically calculates the cross-entropy loss between the ground-truth label and the student's prediction in addition to the KL divergence between the teacher's and student's predictions, this kind of transfer gap makes it ill-prosed to simultaneously align the student's predictive distribution with those mutually exclusive targets, which greatly undermines the power of classical KD.

To get out of this dilemma, this paper rethinks knowledge distillation from a mixture-of-experts (MoE) perspective. The heart of our method, termed MoE-KD, lies in leveraging the teacher's predictions as latent variables to rewrite the classification objective. In this way, we arrive at decomposing the student classifier as a convex combination of conditional models. Namely, each of the conditional models, referred to as an expert, learns to classify a subset of samples, where an input-dependent gating function partitions the dataset into subsets by allocating weights among experts.

To address the nontrivial learning problems, we formulate MoE-KD as an Expectation-Maximization (EM) algorithm (Dempster et al., 1977), where we iteratively estimate the Bayes-optimal posterior distribution of the latent variables given the observed data (in the E-step) and maximize the evidence lower bound (ELBO) of the reformulated classification objective (in the M-step). We theoretically prove that the ELBO is upper-bounded and our proposed EM algorithm contributes to the convergence of the ELBO (see Section 4.3). Empirically, our proposed MoE-KD achieves state-of-the-art performance in various distillation settings regarding teacher-student pairs (homogeneous and heterogeneous) and training datasets (coarse-grained and fine-grained).

## 2 RELATED WORK

### 2.1 KNOWLEDGE DISTILLATION

Knowledge distillation is the process of using a teacher model to improve the performance of a student model. In its classical form, one trains the student to fit the teacher's predictive distribution. (Hinton et al., 2015) popularizes this solution by formulating it as logit matching. MLD (Jin et al., 2023) extends logit matching not only at the instance level but also at the batch and class levels, DKD (Zhao et al., 2022) decouples classical KD into distilling target and non-target class knowledge, and WSLD (Zhou et al., 2021) provides a bias-variance trade-off perspective for the KL term. Besides, the teacher's knowledge can also be distilled in the form of features. One line of feature-based distillation is to mimic the intermediate representations of the teacher network in terms of Euclidean distance (Romero et al., 2014), mutual information (Fu et al., 2023; Tian et al., 2019), Wasserstein distance (Chen et al., 2021a), and maximum mean discrepancy of the network activations (Huang & Wang, 2017) respectively. Another line of feature-based distillation occurs to explore transferring the relationship between features rather than the actual features themselves, where the feature correlation can be captured by the Gram matrix (Yim et al., 2017), Taylor series expansion (Peng et al., 2019), graph (Liu et al., 2019), or quantized visual word space (Jain et al., 2020).

The transfer gap between the teacher and the student is an emerging topic in KD. To mitigate the feature-level transfer gap, MasKD (Huang et al., 2022b) distils the valuable information from receptive regions that contribute to the task precision; NORM (Liu et al., 2023) conducts feature matching in a many-to-one manner, and DiffKD (Huang et al., 2023) explicitly denoises and matches features using diffusion models. When it comes to the logit-level transfer gap, DIST (Huang et al., 2022a) relaxes the KL divergence in logit-based distillation with a correlation-based loss; TAKD (Mirzadeh et al., 2020) introduces multiple middle-sized teaching assistant models to guide the student; DGKD (Son et al., 2021) improves TAKD by densely gathering all the assistant models, and SFTN (Park et al., 2021) provides the teacher with a snapshot of the student during training. Different from these prior works, our method is motivated by the observations that teacher predictions and ground-truth labels indeed behave differently (Niu et al., 2022), arguing that this largely overlooked transfer gap makes it problematic for classical KD to encourage student predictions to simultaneously mimic

the ground-truth labels and teacher predictions. Facing this dilemma, this paper proceeds from a mixture-of-experts perspective by rethinking the role of teacher predictions as latent variables.

## 2.2 MIXTURE OF EXPERTS

The MoE model was initially proposed by (Jacobs et al., 1991) as a technique to combine a series of sub-models and perform conditional computation (Bengio et al., 2015; 2013; Cho & Bengio, 2014) that aims at activating different subsets of a network for different inputs. To increase the model capacity in dealing with complex data, (Ahmed et al., 2016; Gross et al., 2017) extend the MoE structure to the deep neural networks by proposing a deep MoE model composed of multiple layers of routers and experts. Recently, (Shazeer et al., 2017) simplifies the MoE layer by making the output of the gating function sparse for each example, which greatly improves the training stability and reduces the computational cost. Since then, the MoE layer with different base neural network structures has achieved tremendous success in scene parsing (Fu et al., 2018), multi-task learning (Gupta et al., 2022), deep clustering (Chazan et al., 2019; Kopf et al., 2021; Tsai et al., 2021; Zhang et al., 2017), domain generalization (Dai et al., 2021; Li et al., 2022a), data generation (Xia et al., 2022) and question answering (Dai et al., 2022b; Zhou et al., 2022).

KD and MoE tend to evolve mostly independently in the literature. To the best of our knowledge, the only exceptions are (Dai et al., 2022a; Xue et al., 2022). In essence, (Dai et al., 2022a; Xue et al., 2022) exploit the benefits of KD to overcome over-fitting problems (Fedus et al., 2022; Wu et al., 2022) of MoE models on downstream tasks with limited data. On the contrary, this paper formulates the student's classifier as a lightweight MoE layer with the teacher's knowledge to enhance the efficiency of knowledge transfer from the teacher to the student.

## 3 PRELIMINARY

**Notations.** We write vectors and matrices as bold-faced lowercase and uppercase characters respectively. All trainable parameters will be subscripted by $\boldsymbol{\theta}$. Let $\mathbf{e}[i]$ be the $i$-th element of the vector $\mathbf{e} \in \mathbb{R}^K$ and $[K] = \{1, \ldots, K\}$, we then define $\mathrm{softmax}_k(\mathbf{e}) = \exp(\mathbf{e}[k])/\sum_{i \in [K]} \exp(\mathbf{e}[i])$.

**Multi-class Classification.** This paper considers $K$-way classification as a case study, where $\mathcal{X}$ and $\mathcal{Y} = [K]$ denote the input space and label space respectively. Let $\mathbb{P}_{XY}$ be the joint distribution defined over $\mathcal{X} \times \mathcal{Y}$, we are provided with a labelled dataset $\mathcal{D} = \{(\mathbf{x}_1, y_1), \ldots, (\mathbf{x}_N, y_N)\} \sim \mathbb{P}_{XY}^N, i.i.d.$, to train a discriminative model by maximizing the following objective over the dataset $\mathcal{D}$:

$$\mathcal{R}_{\mathrm{cls}}(\mathbf{x}_i, y_i; \boldsymbol{\theta}) \triangleq \log P_{\boldsymbol{\theta}}(Y = y_i | \mathbf{x}_i). \tag{1}$$

**Knowledge Distillation** involves transferring dark knowledge from a teacher model to a student model. Classical KD (Hao et al., 2024; Hinton et al., 2015) calculates the cross-entropy between the ground-truth label and student predictions as well as the KL divergence between the predictive distributions of the student and the teacher. Since the teacher is pre-trained and fixed in the context of KD, the overall learning objective of classical KD can be simplified into the following form[1]:

$$\mathcal{R}_{\mathrm{KD}}(\mathbf{x}_i, y_i; \boldsymbol{\theta}) \triangleq \log P_{\boldsymbol{\theta}}^{\mathcal{S}}(Y^{\mathcal{S}} = y_i | \mathbf{x}_i) + \alpha \cdot \sum_{k=1}^{K} P^{\mathcal{T}}(Y^{\mathcal{T}} = k | \mathbf{x}_i) \log P_{\boldsymbol{\theta}}^{\mathcal{S}}(Y^{\mathcal{S}} = k | \mathbf{x}_i), \tag{2}$$

where $\alpha > 0$ is a weighting hyper-parameter that balances the importance of the two losses. Note that, in Eq. (2), we have used $\mathcal{T}$ and $\mathcal{S}$ as superscripts to indicate the teacher and student model respectively, which, unless explicitly stated, is considered as a default setting in the rest of this paper.

## 4 METHODOLOGY

### 4.1 RETHINKING KNOWLEDGE DISTILLATION: A MIXTURE OF EXPERTS PERSPECTIVE

Motivated by the mixture of experts (MoE) framework (Jacobs et al., 1991), we introduce the teacher's class prediction $Y^{\mathcal{T}} \in \{1, 2, \cdots, K\}$ as a latent variable and naturally extend the vanilla

---

[1]For brevity, we have omitted the constant term $\sum_{k=1}^{K} P^{\mathcal{T}}(Y^{\mathcal{T}} = k | \mathbf{x}_i) \log P^{\mathcal{T}}(Y^{\mathcal{T}} = k | \mathbf{x}_i)$.

classification objective in Eq. (1) to the following formulation:

$$\log P_{\boldsymbol{\theta}}^{\mathcal{S}}(Y^{\mathcal{S}} = y_i | \mathbf{x}_i) = \log \sum_{k=1}^{K} P_{\boldsymbol{\theta}}^{\mathcal{S}} \left( Y^{\mathcal{S}} = y_i, Y^{\mathcal{T}} = k | \mathbf{x}_i \right) \tag{3}$$

$$= \log \underbrace{\sum_{k=1}^{K} P_{\boldsymbol{\theta}}^{\mathcal{S}}(Y^{\mathcal{S}} = y_i | Y^{\mathcal{T}} = k, \mathbf{x}_i) P_{\boldsymbol{\theta}}^{\mathcal{S}} \left( Y^{\mathcal{T}} = k | \mathbf{x}_i \right)}_{\mathcal{R}_{\mathrm{MoE-KD}}(\mathbf{x}_i, y_i; \boldsymbol{\theta})}, \tag{4}$$

where $P_{\boldsymbol{\theta}}^{\mathcal{S}}(Y^{\mathcal{S}} = y_i | Y^{\mathcal{T}} = k, \mathbf{x}_i)$ is one of the experts that classify a subset of samples and $P_{\boldsymbol{\theta}}^{\mathcal{S}} \left( Y^{\mathcal{T}} = k | \mathbf{x}_i \right)$ is a gating function that partitions the dataset into subsets according to the latent semantics by routing each sample to one or a few experts. With this partition-and-classify principle, the experts tend to be highly specialized in data points that share similar semantics, which improves training efficiency. While, same as the original MoE, the experts work in the supervised setting, both gating functions and experts are based on neural networks to fit the high-dimensional data. In the following, we will elaborate on how we parameterize each term in Eq. (4) to fit the KD task.

**Gating function.** The gating function organizes the classification task into $K$ simpler subtasks by weighting the experts based on the semantics of the input sample. Inspired by Du et al. (2017), we formulate the gating function by reusing the pre-trained teacher classifier, namely,

$$P_{\boldsymbol{\theta}}^{\mathcal{S}} \left( Y^{\mathcal{T}} = k | x_i \right) = \mathrm{softmax}_k \left[ g^{\mathcal{T}}(\mathbf{h}_i^{\mathcal{S}}) \right], \quad \mathbf{h}_i^{\mathcal{S}} = \mathcal{G}(\mathbf{z}_i^{\mathcal{S}}), \tag{5}$$

where $\mathbf{z}_i^{\mathcal{S}} \in \mathbb{R}^d$ denotes the student feature of $\mathbf{x}_i$ and a projector $\mathcal{G}(\cdot)$ transforms from the student feature space $\mathcal{Z}^{\mathcal{S}}$ to the teacher feature space $\mathcal{Z}^{\mathcal{T}}$ for dimension alignment at a relatively small cost.

**Experts.** Each expert learns to solve a distinct subtask of the classification task arranged by the gating function. Formally, inspired by Chen et al. (2023), let $\mathbf{e}_k \in \mathbb{R}^d$ be an expert prototype, we formulate the probability of the sample $x_i$ being recognized as the $y_i$-th class by the $k$-th expert as follows:

$$P_{\boldsymbol{\theta}}^{\mathcal{S}} \left( Y^{\mathcal{S}} = y_i | Y^{\mathcal{T}} = k, x_i \right) = \mathrm{softmax}_{y_i} \left[ \mathbf{W}^{\top}(\mathbf{z}_i^{\mathcal{S}} + \mathbf{e}_k) \right] = \mathrm{softmax}_{y_i} \left( \mathbf{W}^{\top} \mathbf{z}_i^{\mathcal{S}} + \mathbf{b}_k \right), \tag{6}$$

where $\mathbf{W} \in \mathbb{R}^{d \times K}$ represents a learnable weight matrix and the expert-specific bias vector $\mathbf{b}_k \in \mathbb{R}^K$ has been re-parameterized by $\mathbf{W}^{\top} \mathbf{e}_k$. To make Eq. (6) benefit from the teacher's logit-level and feature-level knowledge, we implement $\mathbf{e}_k$ based on deep set representations (Zaheer et al., 2017):

$$\mathbf{e}_k = \Psi(\boldsymbol{\mu}_k), \quad \boldsymbol{\mu}_k = \sum_{i=1}^{N} \frac{P^{\mathcal{T}} \left( Y^{\mathcal{T}} = k | \mathbf{x}_i \right)}{\sum_{j=1}^{N} P^{\mathcal{T}} \left( Y^{\mathcal{T}} = k | \mathbf{x}_j \right)} \cdot \mathbf{z}_i^{\mathcal{T}}. \tag{7}$$

where $\Psi(\cdot) : \mathcal{Z}^{\mathcal{T}} \to \mathbb{R}^d$ is another projector that is introduced to match feature dimensions given that Eq. (7) involves a soft aggregation along samples in the teacher embedding space. Although the design of Eq. (7) is similar to pooling by multi-head attention (PMA) in the set transformer (Lee et al., 2019), we do not rely on a softmax operation to normalize aggregation weights along samples as we never expect any single sample to play a dominant role in representing $\boldsymbol{\mu}_k$. Besides, for a fair comparison, we define $P^{\mathcal{T}} \left( Y^{\mathcal{T}} = k | \mathbf{x}_i \right)$ in accordance with prior works (Hao et al., 2024; Hinton et al., 2015; Niu et al., 2022; Zhao et al., 2022; Zhou et al., 2021), i.e.,

$$P^{\mathcal{T}} \left( Y^{\mathcal{T}} = k | \mathbf{x}_i \right) = \mathrm{softmax}_k \left[ g^{\mathcal{T}}(\mathbf{z}_i^{\mathcal{T}}) / \tau \right], \tag{8}$$

where $\tau > 0$ denotes a temperature hyper-parameter (Hinton et al., 2015).

### 4.2 DERIVING THE EVIDENCE LOWER BOUND

In practice, the conditional log-likelihood function in Eq. (4) is hard to be directly optimized (Bishop, 2006; Wang et al., 2021). To address this non-trivial learning problem, let us start from the following evidence lower bound (ELBO) of the vanilla classification objective $\log P_{\boldsymbol{\theta}}^{\mathcal{S}}(Y^{\mathcal{S}} = y_i | \mathbf{x}_i)$:

$$\log P_{\boldsymbol{\theta}}^{\mathcal{S}}(Y^{\mathcal{S}} = y_i | \mathbf{x}_i)$$

$$= \mathrm{ELBO}(\hat{P}, \mathbf{x}_i, y_i; \boldsymbol{\theta}) + D_{\mathrm{KL}} \left[ \hat{P}(Y^{\mathcal{T}} = k | \mathbf{x}_i) || P_{\boldsymbol{\theta}}^{\mathcal{S}}(Y^{\mathcal{T}} = k | Y^{\mathcal{S}} = y_i, \mathbf{x}_i) \right]$$

$$\geq \mathrm{ELBO}(\hat{P}, \mathbf{x}_i, y_i; \boldsymbol{\theta}) \tag{9}$$

$$\triangleq \sum_{k=1}^{K} \left[ \hat{P}(Y^{\mathcal{T}} = k | \mathbf{x}_i) \log P_{\boldsymbol{\theta}}^{\mathcal{S}}(Y^{\mathcal{S}} = y_i | Y^{\mathcal{T}} = k, \mathbf{x}_i) \right] - D_{\mathrm{KL}} \left[ \hat{P}(Y^{\mathcal{T}} = k | \mathbf{x}_i) || P_{\boldsymbol{\theta}}^{\mathcal{S}}(Y^{\mathcal{T}} = k | \mathbf{x}_i) \right],$$

where $D_{\mathrm{KL}}(\cdot)$ represents the Kullback–Leibler (KL) divergence and $\hat{P}(Y^{\mathcal{T}} = k|\mathbf{x}_i)$ denotes any arbitrary distribution conditioned on $\mathbf{x}_i$ such that $\sum_{k=1}^{K} \hat{P}(Y^{\mathcal{T}} = k|\mathbf{x}_i) = 1$. We derive this ELBO in Appendix D to keep the main content concise. To make the inequality hold with equality so that the ELBO reaches its maximum value $\log P_{\boldsymbol{\theta}}^{\mathcal{S}}(Y = y_i|\mathbf{x}_i)$, we need to require:

$$D_{\mathrm{KL}}\left[\hat{P}(Y^{\mathcal{T}} = k|\mathbf{x}_i)||P_{\boldsymbol{\theta}}^{\mathcal{S}}(Y^{\mathcal{T}} = k|Y^{\mathcal{S}} = y_i, \mathbf{x}_i)\right] = 0. \tag{10}$$

By approximating the variational distribution $\hat{P}(Y^{\mathcal{T}} = k|\mathbf{x}_i)$ with $P_{\boldsymbol{\theta}}^{\mathcal{S}}(Y^{\mathcal{T}} = k|Y^{\mathcal{S}} = y_i, \mathbf{x}_i)$ and connecting Eq. (9) with Eq. (4), we are now ready to convert the optimization of Eq. (4) into :

$$\arg\max_{\boldsymbol{\theta}} \mathcal{R}_{\mathrm{MoE-KD}}(\mathbf{x}_i, y_i; \boldsymbol{\theta}) = \arg\max_{\boldsymbol{\theta}} \mathrm{ELBO}(\hat{P}, \mathbf{x}_i, y_i; \boldsymbol{\theta}). \tag{11}$$

## 4.3 Formulating MoE-KD as Expectation-Maximization

**E-step.** This step aims to estimate $\hat{P}_{t+1}(Y^{\mathcal{T}} = k|\mathbf{x}_i)$ with the fixed $\boldsymbol{\theta}_t$ at the iteration $t$ to make $\hat{P}_{t+1}(Y^{\mathcal{T}} = k|\mathbf{x}_i) = P_{\boldsymbol{\theta}_t}^{\mathcal{S}}(Y^{\mathcal{T}} = k|Y^{\mathcal{S}} = y_i, \mathbf{x}_i)$, which is implied by Eq. (10). To this end, by applying the Bayes' theorem to $P_{\boldsymbol{\theta}_t}^{\mathcal{S}}(Y^{\mathcal{T}} = k|Y^{\mathcal{S}} = y_i, \mathbf{x}_i)$, we approximate $\hat{P}_{t+1}(Y^{\mathcal{T}} = k|\mathbf{x}_i)$ as:

$$\hat{P}_{t+1}(Y^{\mathcal{T}} = k|\mathbf{x}_i) = \frac{P_{\boldsymbol{\theta}_t}^{\mathcal{S}}(Y^{\mathcal{T}} = k|\mathbf{x}_i)P_{\boldsymbol{\theta}_t}^{\mathcal{S}}(Y^{\mathcal{S}} = y_i|Y^{\mathcal{T}} = k, \mathbf{x}_i)}{\sum_{j=1}^{K} P_{\boldsymbol{\theta}_t}^{\mathcal{S}}(Y^{\mathcal{T}} = j|\mathbf{x}_i)P_{\boldsymbol{\theta}_t}^{\mathcal{S}}(Y^{\mathcal{S}} = y_i|Y^{\mathcal{T}} = j, \mathbf{x}_i)} \tag{12}$$

**M-step.** With the sub-optimal $\hat{P}_{t+1}(Y^{\mathcal{T}} = k|\mathbf{x}_i) = P_{\boldsymbol{\theta}_t}^{\mathcal{S}}(Y^{\mathcal{T}} = k|Y^{\mathcal{S}} = y_i, \mathbf{x}_i)$ after E-step, we turn to maximize the ELBO in Eq. (9):

$$\boldsymbol{\theta}_{t+1} = \arg\max_{\boldsymbol{\theta}_t} \mathbb{E}_{(\mathbf{x}_i, y_i) \sim \mathbb{P}_{XY}}\left[\mathrm{ELBO}(\hat{P}_{t+1}, \mathbf{x}_i, y_i; \boldsymbol{\theta}_t)\right]. \tag{13}$$

When integrating Eq. (13) into the batch-based training routine where we sample a mini-batch $\mathcal{B}$ from the dataset $\mathcal{D}$ at the beginning of each iteration, it is natural to build an efficient stochastic estimator of the ELBO over $\mathcal{D}$ to learn the parameters $\boldsymbol{\theta}_t$, which is given by:

$$\boldsymbol{\theta}_{t+1} = \arg\max_{\boldsymbol{\theta}_t} \frac{1}{|\mathcal{B}|} \sum_{(\mathbf{x}_i, y_i) \in \mathcal{B}} \mathrm{ELBO}(\hat{P}_{t+1}, \mathbf{x}_i, y_i; \boldsymbol{\theta}_t). \tag{14}$$

**Convergence Analysis.** At the E-step of the iteration $t + 1$, we estimate $\hat{P}_{t+1}(Y^{\mathcal{T}} = k|\mathbf{x}_i)$ to ensure $\mathrm{ELBO}(\hat{P}_{t+1}, \mathbf{x}_i, y_i; \boldsymbol{\theta}_t) = \mathcal{R}_{\mathrm{MoE}}(\mathbf{x}_i, y_i; \boldsymbol{\theta}_t)$. At the M-step after the E-step, we have obtained $\boldsymbol{\theta}_{t+1}$ with a fixed variational distribution $\hat{P}_{t+1}(Y^{\mathcal{T}} = k|\mathbf{x}_i)$, which results in $\mathrm{ELBO}(\hat{P}_{t+1}, \mathbf{x}_i, y_i; \boldsymbol{\theta}_{t+1}) \geq \mathrm{ELBO}(\hat{P}_{t+1}, \mathbf{x}_i, y_i; \boldsymbol{\theta}_t)$. Therefore, we obtain the following sequence:

$$\begin{aligned}
\mathcal{R}_{\mathrm{MoE-KD}}(\mathbf{x}_i, y_i; \boldsymbol{\theta}_{t+1}) &\geq \mathrm{ELBO}(\hat{P}_{t+1}, \mathbf{x}_i, y_i; \boldsymbol{\theta}_{t+1}) \\
&\geq \mathrm{ELBO}(\hat{P}_{t+1}, \mathbf{x}_i, y_i; \boldsymbol{\theta}_t) = \mathcal{R}_{\mathrm{MoE-KD}}(\mathbf{x}_i, y_i; \boldsymbol{\theta}_t).
\end{aligned} \tag{15}$$

Since $\mathcal{R}_{\mathrm{MoE-KD}}(\mathbf{x}_i, y_i; \boldsymbol{\theta}_{t+1}) \geq \mathcal{R}_{\mathrm{MoE-KD}}(\mathbf{x}_i, y_i; \boldsymbol{\theta}_t)$, $\mathrm{ELBO}(\hat{P}, \mathbf{x}_i, y_i; \boldsymbol{\theta})$ is upper-bounded and converge to a certain value with the EM algorithm proposed above. Finally, the inference with the optimized parameters $\hat{\boldsymbol{\theta}}$ for a test-time sample $\mathbf{x}$ requires to compute $\arg\max_k \mathcal{R}_{\mathrm{MoE-KD}}(\mathbf{x}, k; \hat{\boldsymbol{\theta}})$.

## 4.4 Relation to Existing Works

We recently find that SRRL (Yang et al., 2021) also comes with the reused teacher classifier to train the student model and can be regarded as a natural baseline of our method. We forge a mathematical connection between SRRL and the ELBO in Eq. (9) by showing that the latter intrinsically subsumes the former as a special exemplar of itself, which implies the theoretical superiority of our method.

**Assumption 1 (Collapsed Projection)** *The projector $\Psi(\cdot)$ in Eq. (7) is completely collapsed such that, for all inputs $\boldsymbol{\mu} \in \mathcal{Z}^{\mathcal{T}}$, we have $\Psi(\boldsymbol{\mu}) = \mathbf{b}$.*

**Lemma 1** *If Assumption 1 holds, the expert $P_{\boldsymbol{\theta}}^{\mathcal{S}}\left(Y^{\mathcal{S}} = y_i|Y^{\mathcal{T}} = k, x_i\right)$ in Eq. (6) will degenerate into a universal parametric softmax classifier, which is given by:*

$$P_{\boldsymbol{\theta}}^{\mathcal{S}}\left(Y^{\mathcal{S}} = y_i|Y^{\mathcal{T}} = k, x_i\right) = \mathrm{softmax}_{y_i}\left[\mathbf{W}^{\top}\mathbf{z}_i^{\mathcal{S}} + \mathbf{b}\right], \quad \forall k \in [K]. \tag{16}$$

Table 1: Top-1 ACC (%) on CIFAR-100, Homogenous Architecture. The best results are in boldface.

| Teacher | WRN-40-2 | WRN-40-2 | ResNet56 | ResNet110 | ResNet32x4 | VGG13 |
|---|---|---|---|---|---|---|
| | 75.61 | 75.61 | 72.34 | 74.31 | 79.42 | 74.64 |
| Student | WRN-16-2 | WRN-40-1 | ResNet20 | ResNet32 | ResNet8x4 | VGG8 |
| | 73.26 | 71.98 | 69.06 | 71.14 | 72.50 | 70.36 |
| KD | 74.92 | 73.54 | 70.66 | 73.08 | 73.33 | 72.98 |
| FitNet | 73.58 | 72.24 | 69.21 | 71.06 | 73.50 | 71.02 |
| CRD | 75.48 | 74.14 | 71.16 | 73.48 | 75.51 | 73.94 |
| WCoRD | 75.88 | 74.73 | 71.56 | 73.81 | 75.95 | 74.55 |
| IPWD | — | 74.64 | 71.32 | 73.91 | 76.03 | — |
| WSLD | — | 74.48 | 72.15 | 74.12 | 76.05 | — |
| SRRL | 75.96 | 74.75 | 71.44 | 73.80 | 75.92 | 74.40 |
| DKD | 76.24 | 74.81 | 71.97 | 74.11 | 76.32 | 74.68 |
| NORM | 75.65 | 74.82 | 71.35 | 73.67 | 76.49 | 73.95 |
| DIST | — | 74.73 | 71.75 | — | 76.31 | — |
| DiffKD | — | 74.09 | 71.92 | — | 76.72 | — |
| WTTM | 76.37 | 74.58 | 71.92 | 74.13 | 76.06 | 74.44 |
| Ours | **76.98** | **75.21** | **72.49** | **74.58** | **77.10** | **75.03** |

In addition to Assumption 1, let $\hat{P}(Y^{\mathcal{T}} = k|\mathbf{x}_i) = P^{\mathcal{T}}(Y^{\mathcal{T}} = k|\mathbf{x}_i)$, the ELBO in Eq. (9), i.e.,

$$\text{ELBO}(\hat{P}, \mathbf{x}_i, y_i; \boldsymbol{\theta}) = \underbrace{\log\left[\text{softmax}_{y_i}\left(\mathbf{W}^\top \mathbf{z}_i^{\mathcal{S}} + \mathbf{b}\right)\right] - D_{\text{KL}}\left[P^{\mathcal{T}}(Y^{\mathcal{T}} = k|\mathbf{x}_i)\|P_{\boldsymbol{\theta}}^{\mathcal{S}}(Y^{\mathcal{T}} = k|\mathbf{x}_i)\right]}_{\mathcal{R}_{\text{SRRL}}(\mathbf{x}_i, y_i; \boldsymbol{\theta})},$$

is mathematically equivalent to the optimization objective in SRRL regardless the hyper-parameter $\beta$ that scales the second term of $\mathcal{R}_{\text{SRRL}}(\mathbf{x}_i, y_i; \boldsymbol{\theta})$. Nevertheless, it is worthwhile to point out that, as disclosed by the authors of SRRL, $\beta = 1$ contributes to the best knowledge transfer performance, which empirically epochs our analysis above.

Interestingly, if we treat ground-truth annotations as a noisy version of teacher predictions and the gating function in Eq. (4) as the clean class posterior, the experts in Eq. (4) share a similar working mechanism with the so-called noise transition matrix $\mathbf{T} \in [0,1]^{K \times K}$ (Patrini et al., 2017) in label-noise learning (Song et al., 2022) such that $\mathbf{T}_{ij}(\mathbf{x}) = P_{\boldsymbol{\theta}}^{\mathcal{S}}(Y^{\mathcal{S}} = i|Y^{\mathcal{T}} = j, \mathbf{x})$. However, directly estimating the transition matrix is generally infeasible (Xia et al., 2019) without the rigorous anchor-point assumption (Liu & Tao, 2015). As a result, existing label-noise learners (Cheng et al., 2022a; Xia et al., 2020; Yang et al., 2022) have been developed with a two-stage training routine: 1) pre-training the gating function to estimate experts (or the noise transition matrix $\mathbf{T}$) and 2) fine-tuning the gating function with the fixed estimated experts. By contrast, our method enables to simultaneously learn both the gating function and experts. While the mostly recent works (Cheng et al., 2022b; Li et al., 2021) approach label-noise learning in an end-to-end way, they can be criticized for removing the dependency between $P_{\boldsymbol{\theta}}^{\mathcal{S}}(Y^{\mathcal{S}} = i|Y^{\mathcal{T}} = j, \mathbf{x})$ and $\mathbf{x}$ to have

$$P_{\boldsymbol{\theta}}^{\mathcal{S}}(Y^{\mathcal{S}} = i|Y^{\mathcal{T}} = j, \mathbf{x}) = P_{\boldsymbol{\theta}}^{\mathcal{S}}(Y^{\mathcal{S}} = i|Y^{\mathcal{T}} = j), \quad \forall i, j \in [K]. \tag{17}$$

## 5 EXPERIMENTS

**Datasets.** We perform experiments on CIFAR-100 (Krizhevsky et al., 2009), ImageNet-1K (Russakovsky et al., 2015), Tiny-ImageNet (Tavanaei, 2020), STL-10 (Coates et al., 2011), CUB (Wah et al., 2011) and Stanford Dogs (Khosla et al., 2011), following prior works (Huang et al., 2023; Li et al., 2022b; Zhao et al., 2022).

**Baselines.** We compare our method with advanced methods including KD (Hinton et al., 2015), DKD (Zhao et al., 2022), IPWD (Niu et al., 2022), WSLD (Zhou et al., 2021), ESKD (Cho & Hariharan, 2019), TAKD (Mirzadeh et al., 2020), SCKD (Zhu & Wang, 2021), NKD (Yang et al., 2023), DIST (Huang et al., 2022a), FitNets (Romero et al., 2014), CRD (Tian et al., 2019), WCoRD (Chen et al., 2021a), ReviewKD (Chen et al., 2021b), NORM (Liu et al., 2023), DiffKD (Huang et al., 2023),

Table 2: Top-1 ACC (%) on CIFAR-100, Heterogeneous Architecture. The best result is in boldface.

| Teacher | VGG13 | ResNet50 | ResNet32x4 | ResNet32x4 | WRN-40-2 |
|---|---|---|---|---|---|
| | 74.64 | 79.34 | 79.42 | 79.42 | 75.61 |
| Student | MobileNetV2 | MobileNetV2 | ShuffleNetV1 | ShuffleNetV2 | ShuffleNetV1 |
| | 64.60 | 64.60 | 70.50 | 71.82 | 70.50 |
| KD | 67.37 | 67.35 | 74.07 | 74.45 | 74.83 |
| FitNet | 64.14 | 63.16 | 73.59 | 73.54 | 73.73 |
| CRD | 69.73 | 69.11 | 75.11 | 75.65 | 76.05 |
| WCoRD | 69.47 | 70.45 | 75.40 | 75.96 | 76.32 |
| IPWD | — | 70.25 | 76.03 | — | 76.44 |
| WSLD | — | — | 75.46 | 75.93 | 76.21 |
| SRRL | 69.14 | 69.45 | 75.66 | 76.40 | 76.61 |
| DKD | 69.71 | 70.35 | 76.45 | 77.07 | 76.70 |
| NORM | 68.94 | 70.56 | 77.42 | 78.07 | 77.06 |
| DIST | — | 68.66 | 76.34 | 77.35 | — |
| DiffKD | — | 69.21 | 76.57 | 77.52 | — |
| WTTM | 69.16 | 69.59 | 74.37 | 76.55 | 75.42 |
| SKD | 68.79 | 69.55 | — | 76.67 | 76.65 |
| Ours | **70.54** | **71.38** | **78.23** | **78.69** | **77.52** |

ITRD (Miles et al., 2021), SRRL (Yang et al., 2021), WTTM (Zheng & YANG, 2024), LSKD (Sun et al., 2024), and SKD (Wei et al., 2024).

**Settings.** We employ the last feature map and a three-layer bottleneck transformation for implementing the projector $\mathcal{G}(\cdot)$, which only incurs a less than 3% cost to the pruning ratio in teacher-to-student compression (Chen et al., 2022). We design $\Psi(\cdot)$ as a two-layer MLP module (Chen et al., 2020). As for the temperature $\tau$, the *only* hyper-parameter in our method, we empirically find that the common setting, i.e., $\tau = 4$ for CIFAR-100 and $\tau = 1$ for ImageNet-1K, is sufficient to achieve satisfactory performance. The reported results of our method are averaged over 5 runs.

## 5.1 MAIN RESULTS

**CIFAR-100.** To evaluate the effectiveness of our method, we experiment on CIFAR100 with 11 student-teacher combinations. We consider a standard data augmentation scheme including padding 4 pixels before random cropping and horizontal flipping. We set the batch size as 64 and the initial learning rate as 0.01 (for ShuffleNet and MobileNet-V2) or 0.05 (for the other series). We train the model for 240 epochs, in which the learning rate is decayed by 10 every 30 epochs after 150 epochs. We use SGD as the optimizer with weight decay $5e-4$ and momentum 0.9, Table 1 and Table 2 compare the Top-1 accuracy under two different scenarios respectively: 1) the student and the teacher share the same network architecture and 2) the student and the teacher are of a different architectural style. The results show that ours surpasses previous methods in all cases. Taking the ResNet32x4/ResNet8x4 and WRN-40-2/ShuffleNetV1 pairs as an example, our method outperforms the most recent WTTM by 1.04% and 2.10% for each.

**ImageNet-1K.** To validate the scalability of our method, we employ the PyTorch-version student-teacher combinations to perform experiments on ImageNet. The standard PyTorch ImageNet practice is adopted except for 100 training epochs. We set the batch size as 256 and the initial learning rate as 0.1. The learning rate is divided by 10 for every 30 epochs. We use SGD as the optimizer with weight decay $1e-4$ and momentum 0.9. The Top-1 and Top-5 accuracy of different distillation methods are reported in Table 3. While our method slightly performs worse than the state-of-the-art DiffKD by 0.21% for the ResNet50/MobileNetV1 pair, we achieve significantly better performance than DiffKD by 0.82% and 0.64% for the ResNet34/ResNet18 pair regarding Top-1 and Top-5 accuracy.

## 5.2 ABLATION STUDY

We conduct an ablation study to validate our motivation and design, with the following baselines.

Table 3: Top-1 and Top-5 ACC (%) on ImageNet-1K. The best result is in boldface.

| Teacher: ResNet34 → Student: ResNet18 | | | Teacher: ResNet50 → Student: MobileNetV1 | | |
|---|---|---|---|---|---|
| Method | Top-1 ACC | Top-5 ACC | Method | Top-1 ACC | Top-5 ACC |
| Teacher | 73.31 | 91.42 | Teacher | 76.16 | 92.87 |
| Student | 69.75 | 89.07 | Student | 68.87 | 88.76 |
| KD | 70.66 | 89.88 | KD | 70.68 | 90.30 |
| WSLD | 72.04 | 90.70 | WSLD | 71.52 | 90.34 |
| NKD | 71.96 | 90.48 | NKD | 72.58 | 90.96 |
| DKD | 71.70 | 90.41 | DKD | 72.05 | 91.05 |
| DIST | 72.07 | 90.42 | DIST | 73.24 | 91.12 |
| CRD | 71.17 | 90.13 | CRD | 71.31 | 90.41 |
| ReviewKD | 71.61 | 90.51 | ReviewKD | 72.56 | 91.00 |
| DiffKD | 72.22 | 90.64 | DiffKD | **73.62** | 91.34 |
| SRRL | 71.73 | 90.60 | SRRL | 72.49 | 90.92 |
| WTTM | 72.19 | — | WTTM | 73.09 | — |
| LSKD | 71.42 | 90.29 | LSKD | 72.18 | 90.80 |
| Ours | **73.15** | **92.28** | Ours | 73.41 | **91.35** |

Table 4: Ablation study results on CIFAR-100. Each row shows the Top-1 ACC (%).

| Teacher → Student | Baseline (i) | Baseline (ii) | Baseline (iii) | Baseline (iv) | Baseline (v) | Full model |
|---|---|---|---|---|---|---|
| WRN-40-2 → WRN-40-1 | 73.87 | 74.43 | 74.66 | 74.95 | 74.79 | 75.21 |
| ResNet50 → MobileNetV2 | 69.28 | 70.62 | 70.65 | 71.16 | 70.94 | 71.38 |

 (i) We validate the necessity of the gating function in our method by simplifying the gating function as a uniform one such that $P_{\boldsymbol{\theta}}^{\mathcal{S}}\left(Y^{\mathcal{T}} = k|\mathbf{x}_i\right) = 1/K$.

 (ii) We justify the use of the teacher's classifier for the gating function by learning the gating function from scratch with the student.

(iii) In analogy to (ii), we learn the subtask-specific embedding vector from scratch with the student.

(iv) We replace the soft aggregation strategy in Eq. (7) with the hard aggregation strategy based on the hard assignments produced by the teacher.

 (v) As described in Section 4.3, we formulate our method as an EM algorithm where we keep estimating the Bayes-optimal variational distribution. In this baseline, we replace the Bayes-optimal estimation with the direct assignment of the teacher's predictive distribution.

**Baseline Comparison.** Experimental results of the ablation study on CIFAR-100 are shown in Table 4. We note several interesting observations: 1) The performance drop in Baselines (ii) and (iii) show that introducing the teacher's knowledge from either outputs or architecture is beneficial to the student; 2) Baseline (iv) performs worse than the full model. An explanation is that, compared with the hard aggregation in Baseline (iv), the soft aggregation in the full model injects class relationship knowledge so that smoothness between classes is preserved in subtask-specific embedding for each expert; 3) Baseline (i) performs worst among the baselines, which could be attributed to that, with a uniform prior, the experts would take extra efforts to become specialized in a set of images with shared semantics; 4) Baseline (v) also performs worse than the full model, which implies that it is non-trivial to properly estimate the variational distribution.

## 5.3 EXTENTIONS

**Feature Transferability.** To study the generalization of our method, we evaluate our distilled model on downstream tasks. In particular, we employ linear probing on STL-10 and Tiny-ImageNet. We freeze the student model and train a linear classifier on the top of the student backbone to perform 10-way and 200-way classification for STL-10 and Tiny-ImageNet (all images down-sampled to $32 \times 32$). More implementation details are attached in Appendix B. Our results in Table 5 indicate the superior transferability of features learned by our method.

Table 5: Linear probing on STL-10 and Tiny-ImageNet: We use the combination of teacher WRN-40-2 and student WRN-16-2. We report Top-1 ACC (%). The best result is in boldface.

| Source → Target | Student | KD | DKD | FitNet | ReviewKD | CRD | ITRD | Ours |
|---|---|---|---|---|---|---|---|---|
| CIFAR-100 → STL-10 | 69.7 | 70.9 | 72.9 | 70.3 | 72.4 | 71.6 | 72.7 | **73.4** |
| CIFAR-100 → Tiny-ImageNet | 33.7 | 33.9 | 37.1 | 33.5 | 36.6 | 35.6 | 36.0 | **37.5** |

Table 6: Top-1 ACC (%) on CUB and Stanford Dogs compared to advanced knowledge distillers. We use the ResNeXt101-32-8d as the teacher for both datasets. The best result is shown in boldface.

| Datasets | CUB | | | Stanford Dogs | | |
|---|---|---|---|---|---|---|
| Student | AlexNet | ShuffleNetV2 | MobileNetV2 | AlexNet | ShuffleNetV2 | MobileNetV2 |
| Random Init. | 55.66 | 71.24 | 74.49 | 50.20 | 68.72 | 68.67 |
| KD | 55.10 | 71.89 | 76.45 | 50.22 | 68.48 | 71.25 |
| ESKD | 55.64 | 72.15 | 76.87 | 50.39 | 69.02 | 71.56 |
| TAKD | 54.82 | 71.53 | 76.25 | 50.36 | 68.94 | 70.61 |
| SCKD | 56.78 | 71.99 | 75.13 | 51.78 | 68.80 | 70.13 |
| KD+ATS | 58.32 | 73.15 | 77.83 | 52.96 | 70.92 | 73.16 |
| Ours | **59.46** | **74.09** | **78.68** | **53.85** | **72.15** | **74.23** |

Table 7: Top-1 ACC (%) on ImageNet-1K with ResNet50 trained by Wightman et al. (2021) as a stronger teacher. Students are trained under a stronger strategy (Huang et al., 2022a; 2023).

| Teacher → Student | Random Init. | KD | RKD | SRRL | DIST | DiffKD | Ours |
|---|---|---|---|---|---|---|---|
| ResNet50 → ResNet34 | 76.8 | 77.2 | 76.6 | 76.7 | 77.8 | 78.1 | **78.3** |
| ResNet50 → EfficientNet-B0 | 78.0 | 77.4 | 77.5 | 77.3 | 78.6 | 78.8 | **79.0** |

**Fine-grained Classification.** In practice, available training samples may be visually similar. We validate our method in the scenario of fine-grained classification. Table 6 reports the Top-1 accuracy of state-of-the-art methods on CUB and Stanford Dogs. It can be found that KD can help to improve the performance of a student network. The improvement can be further enhanced by the early-stopped teacher in ESKD (Bengio et al., 2013), the teacher assistant in TAKD (Cho & Bengio, 2014), the student-customized teacher in SCKD (Shazeer et al., 2017) and the asymmetric temperature scaling in ATS (Li et al., 2022b). Nevertheless, our method consistently contributes to the most significant improvement for various student networks.

**Distillation with Stronger Teachers.** To fully investigate the sensitivity of MoE-KD to the capacity gap between the teacher and the student, we further conduct experiments on teachers with stronger training strategies following DIST. It can be observed from Table 7 that our method keeps achieving the best performance for both ResNet50/ResNet34 and ResNet50/EfficientNet-B0 pairs. In particular, while the state-of-the-art DiffKD improves Top-1 accuracy by 1.3% and 0.8% for ResNet34 and EfficientNet-B0 respectively, our proposed method enhances the two by 1.5% and 1.0%.

## 6 CONCLUSION

In this paper, we study knowledge distillation from a novel mixture-of-experts perspective, where we leverage the teacher's knowledge from outputs and learned parameters to tackle the classification within a partition-and-classify principle. Our method comprises an input-dependent gating function that distributes subtasks to one or a few specialized experts, and multiple experts that classify the subset of samples based on sample-level student representations and class-level teacher representations. Moreover, by deriving the ELBO, our model can be formulated as an expectation-maximization algorithm and trained without requiring any other loss or regularization terms. Extensive experiments show that our method is empirically effective in not only consistently boosting the student model classification performance in various distillation settings but also improving the feature transferability.

ETHIC IMPACTS

Investigating the efficacy of the proposed method would consume considerable computing resources. These efforts can contribute to increased carbon emissions, which could raise environmental concerns. This paper does not raise any more ethical concerns due to the un-involvement of any human subjects' practices to data set releases, potentially harmful insights, methodologies and applications, potential conflicts of interest and sponsorship, discrimination/bias/fairness concerns, privacy and security issues, legal compliance, and research integrity issues.

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

## A   LIMITATIONS.

This paper only explores one of the parameterization schemes for the proposed mixture-of-experts framework. It will be exciting to explore more possibilities for parameterization in the future when promoting KD from the mixture-of-experts perspective.

## B   IMPLEMENTATION DETAILS FOR LINEAR PROBING

We utilize an SGD optimizer with a momentum of 0.9, a batch size of 64 and a weight decay of 0. The initial learning rate starts at 0.1 and is decayed by 10 at the 30-th, 60-th and 90-th epochs within a total of 100 epochs.

## C   STANDARD DEVIATION FOR THE REPORTED RESULTS ON CIFAR-100

Below, we report the standard deviation (Std.) for the experiment results of our method on CIFAR-100 in Table 8 and Table 9. Results are averaged over 5 independent runs.

Table 8: Top-1 ACC (%) on CIFAR-100, Homogenous Architecture. The best results are in boldface.

| Teacher | WRN-40-2 | WRN-40-2 | ResNet56 | ResNet110 | ResNet32x4 | VGG13 |
|---------|----------|----------|----------|-----------|------------|-------|
| Student | WRN-16-2 | WRN-40-1 | ResNet20 | ResNet32  | ResNet8x4  | VGG8  |
| Mean    | 76.98    | 75.21    | 72.49    | 74.58     | 77.10      | 75.03 |
| Std.    | 0.37     | 0.32     | 0.26     | 0.27      | 0.41       | 0.25  |

Table 9: Top-1 ACC (%) on CIFAR-100, Heterogeneous Architecture. The best result is in boldface.

| Teacher | VGG13 | ResNet50 | ResNet32x4 | ResNet32x4 | WRN-40-2 |
|---------|-------|----------|------------|------------|----------|
| Student | MobileNetV2 | MobileNetV2 | ShuffleNetV1 | ShuffleNetV2 | ShuffleNetV1 |
| Mean    | 70.54 | 71.38    | 78.23      | 78.69      | 77.52    |
| Std.    | 0.46  | 0.29     | 0.24       | 0.43       | 0.35     |

## D   THE ELBO DERIVATION

To begin with, we formally state the facts that will be used in our derivation:

**Fact 1**. Since $Y^{\mathcal{T}} \in \{1, 2, \cdots, K\}$, for any arbitrary distribution $\hat{P}(Y^{\mathcal{T}} = k|\mathbf{x}_i)$, we have $\sum_{k=1}^{K} \hat{P}(Y^{\mathcal{T}} = k|\mathbf{x}_i) \equiv 1$

**Fact 2**. For the events A, B, C, Bayes' theorem implies that $P(A, C|B) = P(A|B, C)P(C|B)$

**Fact 3**. Based on Fact 2, for the events A, B, C, we have $P(C|B) = \frac{P(A,C|B)}{P(A|B,C)}$

**Fact 4**. Since $Y^{\mathcal{T}} \in \{1, 2, \cdots, K\}$, for any arbitrary distributions $\hat{P}(Y^{\mathcal{T}} = k|\mathbf{x}_i)$ and $P(Y^{\mathcal{T}} = k|\mathbf{x}_i)$, we have $D_{\mathrm{KL}}\left[\hat{P}(Y^{\mathcal{T}} = k|\mathbf{x}_i)||P(Y^{\mathcal{T}} = k|\mathbf{x}_i)\right] \geq 0$.

$$
\begin{aligned}
&\log P_{\boldsymbol{\theta}}^{\mathcal{S}}(Y^{\mathcal{S}} = y_i|\mathbf{x}_i) \\
=&1 \cdot \log P_{\boldsymbol{\theta}}^{\mathcal{S}}(Y^{\mathcal{S}} = y_i|\mathbf{x}_i) \\
=&\left[\sum_{k=1}^{K} \hat{P}(Y^{\mathcal{T}} = k|\mathbf{x}_i)\right] \cdot \log P_{\boldsymbol{\theta}}^{\mathcal{S}}(Y^{\mathcal{S}} = y_i|\mathbf{x}_i) \quad \textbf{(Fact 1)} \\
=&\sum_{k=1}^{K} \left[\hat{P}(Y^{\mathcal{T}} = k|\mathbf{x}_i) \log P_{\boldsymbol{\theta}}^{\mathcal{S}}(Y^{\mathcal{S}} = y_i|\mathbf{x}_i)\right] \\
=&\sum_{k=1}^{K} \hat{P}(Y^{\mathcal{T}} = k|\mathbf{x}_i) \log \frac{P_{\boldsymbol{\theta}}^{\mathcal{S}}(Y^{\mathcal{S}} = y_i, Y^{\mathcal{T}} = k|\mathbf{x}_i)}{P_{\boldsymbol{\theta}}^{\mathcal{S}}(Y^{\mathcal{T}} = k|Y^{\mathcal{S}} = y_i, \mathbf{x}_i)}, \quad \textbf{(Fact 3)} \\
=&\sum_{k=1}^{K} \hat{P}(Y^{\mathcal{T}} = k|\mathbf{x}_i) \log \frac{P_{\boldsymbol{\theta}}^{\mathcal{S}}(Y^{\mathcal{S}} = y_i|\mathbf{x}_i, Y^{\mathcal{T}} = k)P_{\boldsymbol{\theta}}^{\mathcal{S}}(Y^{\mathcal{T}} = k|\mathbf{x}_i)}{P_{\boldsymbol{\theta}}^{\mathcal{S}}(Y^{\mathcal{T}} = k|Y^{\mathcal{S}} = y_i, \mathbf{x}_i)} \quad \textbf{(Fact 2)} \\
=&\sum_{k=1}^{K} \hat{P}(Y^{\mathcal{T}} = k|\mathbf{x}_i) \log \left[\frac{P_{\boldsymbol{\theta}}^{\mathcal{S}}(Y^{\mathcal{S}} = y_i|\mathbf{x}_i, Y^{\mathcal{T}} = k)P_{\boldsymbol{\theta}}^{\mathcal{S}}(Y^{\mathcal{T}} = k|\mathbf{x}_i)}{P_{\boldsymbol{\theta}}^{\mathcal{S}}(Y^{\mathcal{T}} = k|Y^{\mathcal{S}} = y_i, \mathbf{x}_i)} \cdot \frac{\hat{P}(Y^{\mathcal{T}} = k|\mathbf{x}_i)}{\hat{P}(Y^{\mathcal{T}} = k|\mathbf{x}_i)}\right] \\
=&\sum_{k=1}^{K} \left[\hat{P}(Y^{\mathcal{T}} = k|\mathbf{x}_i) \log \frac{P_{\boldsymbol{\theta}}^{\mathcal{S}}(Y^{\mathcal{S}} = y_i|\mathbf{x}_i, Y^{\mathcal{T}} = k)P_{\boldsymbol{\theta}}^{\mathcal{S}}(Y^{\mathcal{T}} = k|\mathbf{x}_i)}{\hat{P}(Y^{\mathcal{T}} = k|\mathbf{x}_i)}\right] \\
&\hspace{5cm} + D_{\mathrm{KL}}\left[\hat{P}(Y^{\mathcal{T}} = k|\mathbf{x}_i)||P_{\boldsymbol{\theta}}^{\mathcal{S}}(Y^{\mathcal{T}} = k|Y^{\mathcal{S}} = y_i, \mathbf{x}_i)\right] \\
\geq&\sum_{k=1}^{K} \hat{P}(Y^{\mathcal{T}} = k|\mathbf{x}_i) \log \frac{P_{\boldsymbol{\theta}}^{\mathcal{S}}(Y^{\mathcal{S}} = y_i|Y^{\mathcal{T}} = k, \mathbf{x}_i)P_{\boldsymbol{\theta}}^{\mathcal{S}}(Y^{\mathcal{T}} = k|\mathbf{x}_i)}{\hat{P}(Y^{\mathcal{T}} = k|\mathbf{x}_i)} \quad \textbf{(Fact 4)} \\
=&\sum_{k=1}^{K} \left[\hat{P}(Y^{\mathcal{T}} = k|\mathbf{x}_i) \log P_{\boldsymbol{\theta}}^{\mathcal{S}}(Y^{\mathcal{S}} = y_i|Y^{\mathcal{T}} = k, \mathbf{x}_i)\right] - D_{\mathrm{KL}}\left[\hat{P}(Y^{\mathcal{T}} = k|\mathbf{x}_i)||P_{\boldsymbol{\theta}}^{\mathcal{S}}(Y^{\mathcal{T}} = k|\mathbf{x}_i)\right] \\
=&\mathrm{ELBO}(\hat{P}, \mathbf{x}_i, y_i; \boldsymbol{\theta}).
\end{aligned}
$$

$$(18)$$

