# OpenReview forum: "Rethinking Knowledge Distillation: A Mixture-of-Experts Perspective"
_ICLR.cc/2025/Conference — Submitted to ICLR 2025_

### Official Review · Reviewer_xfaT · 2024-11-02

**Soundness:** 3
**Presentation:** 3
**Contribution:** 3
**Rating:** 5
**Confidence:** 4

**Summary:**

This research investigates the process of knowledge distillation from a mixture-of-experts perspective. The authors redefine the distillation objective by incorporating teacher predictions as latent variables and introduce MoE-KD to deconstruct the vanilla classification task into a blend of simpler subtasks with the teacher classifier acting as a gating mechanism to assess the significance of the subtasks. Experimental results indicate that the proposed MoE-KD, functioning as an Expectation-Maximization algorithm, surpasses sophisticated knowledge distillation techniques across various benchmarks.

**Strengths:**

1. The concept of integrating teacher predictions as latent variables and redefining the distillation objective using the MoE framework is intriguing and commendable.

2. The experiments consistently show that the proposed MoE-KD outperforms existing distillation methods.

**Weaknesses:**

1. As the proposed method is formulated as an Expectation-Maximization algorithm, the computation cost of the proposed method seems much more expensive than existing methods. It would be better if the authors could provide the computation comparisons in the experiments.

2. It seems the term P^S_{theta}(Y^T=k|x_i) in Eqn. (12) intractable as the predictions of the teacher could not be derived by the student. How the authors address the problem? (If some misunderstandings exist, please correct me).

3. It would be better if the authors could provide one diagram or pseudo code of the whole algorithm to illustrate the framework of proposed method. In the current status, it seems a bit confusing how the student model in the experiments is adapted to a MoE version, and, after training, how the student model makes inference without the teacher predictions.

4. The experiments are conducted on only CNN networks (both the teachers and the students). I wonder how such a method performs with vision transformers.

5. Some highly related works are missing in the related work. For example, Xue et al., [1] also propose use a Mixture-of-Experts model to explain knowledge distillation. What are the differences between their work and the proposed method? The authors should include discussions and comparison with such highly related work.

[1] KDExplainer: A Task-oriented Attention Model for Explaining Knowledge Distillation. IJCAI 2021

**Questions:**

Please see the Weaknesses

---

### Official Review · Reviewer_dbcv · 2024-11-03

**Soundness:** 3
**Presentation:** 2
**Contribution:** 3
**Rating:** 5
**Confidence:** 3

**Summary:**

This paper proposes a new knowledge distillation method named MoE-KD, which partitions the predicted logits into multiple groups from a mixture-of-experts perspective, and then conduct distillation separately in each group. Then, by deriving the evidence lower bound of classification objective, the authors further formulate the distillation objective as an expectation-maximization algorithm to adress the nontrival learning problems in the vanilla log-likelihood objective. Experiments on CIFAR-100, ImageNet, and various transfer learning datasets are conducted.

**Strengths:**

1. This paper proposes a novel perspective of knowledge distillation by splitting the classification into multiple subtasks, which sounds interesting and effective to solve the distribution imbalance problem occured in multi-label classification distillation.

2. The authors conduct extensive experiments to validate the effectiveness. The proposed method outperforms previous KD methods in almost all the experimented settings. Notably, when with the latest strong training strategy and large teacher-student gap, the method can still achieve the optimal performance.

3. The paper gives sufficient comparisons to existing methods, and clearly elaborates the novelty of the method.

**Weaknesses:**

1. The methodology section may be difficult to follow and the readibility can be improved. The transition from Section 4.2 to 4.3 feels abrupt, and the shifts from EBLO maximization to EM algorithm lacks a clear, guiding explanation that ties the two subsections.

2. The ablation study in Table 4 is insufficient. It evaluates the impact of removing specific components from the complete method to demonstrate the effectivenesses. However, it does not provide insight into the standalone effectiveness when applied each to the KD baseline alone, if feasible. For example, one would like to see the effect of using gating function to partition the task into subtasks in KD baseline.

3. It would be better to design some toy experiments or show some visualizations to demonstrate the influence of the method, rather than simply comparing the final accuracies. For example, visualize the distribution of gates; how is the sharpness of the distribution? Any insights from the learned distributions?

**Questions:**

4. What is the value of $K$ used in the experiments? Will the choices of $K$ affect the performance?
5. In Section 5.3, any explanations on why the proposed method has better transferability?
6. Is the proposed method applicable to feature-level distillation, as feature distillation is important for dense prediction tasks such as object detection?

---

### Official Review · Reviewer_QXWV · 2024-11-04

**Soundness:** 3
**Presentation:** 2
**Contribution:** 3
**Rating:** 5
**Confidence:** 4

**Summary:**

The authors raise an issue in knowledge distillation regarding cases where the ground truth and the teacher’s prediction do not align. To address this, they propose viewing the existing classification objective as a latent MoE (Mixture of Experts) from the teacher's perspective and suggest an optimization method based on the EM (Expectation-Maximization) algorithm.

**Strengths:**

- It is interesting to interpret the existing objective from the perspective of the teacher as a latent variable.
- They provide mathematical explanation of MoE-KD.

**Weaknesses:**

- The consistency between the problem statement, the proposed algorithm, and the problem-solving approach is lacking.
  - How frequently do cases occur in the training data samples where there is a significant discrepancy between the ground truth and the teacher’s prediction?
  - From what perspective is the proposed method better than the traditional direct KL divergence approach? Does it help facilitate learning toward the ground truth more effectively when there is a significant difference between the ground truth and the teacher’s prediction?
  - It would be beneficial to show the training results using only data samples with a large gap between ground truth and the teacher's prediction.

**Questions:**

- (writing) Please distinguish between \citep and \citet for proper citations.
- It would be helpful if the relevance to existing knowledge distillation methods were clarified.

---

> ### Comment · Reviewer_QXWV · 2024-12-01
>
> As the response (Q1/Q3) appears unclear, I will keep the current score as it is.
>
> For Q1, it is necessary to first explain how "discrepancy" is defined. Additionally, the question pertains to a specific teacher-student scenario, for example, with CIFAR-100. It asks about the proportion of the 50,000 training set samples that exhibit a discrepancy between the ground truth and the teacher.
>
> For Q3, it is necessary to identify the differences between the existing methods and the proposed method when applied to the specific dataset obtained above.

---

### Official Review · Reviewer_SXc3 · 2024-11-04

**Soundness:** 3
**Presentation:** 2
**Contribution:** 3
**Rating:** 5
**Confidence:** 4

**Summary:**

The paper proposes an interesting re-formulation of the knowledge distillation setup from a mixture of experts perspective by putting up the arguement that the distillation from teacher might not be well aligned from the classification objective under some circumstances. Therefore, they treat the teacher prediction label as a latent variable and instead rewrite the classification objective in form of a mixture of experts setup where the prediction conditioned on a particular value of the teacher prediction and the input is treated as one of the experts that classify one of the samples and the prediction that  a given input results in a particular value of teacher label acts as a gating function. Both these are parameterized as neural networks where the gating function is just the teacher's classifier on top of student extracted features and a transformation to the feature space of teacher. Furthermore, they formulate the probability of a sample being recognized as a particular class by a particular expert as a transformation of student feature via a learnable weight matrix and a bias depending to this and expert prototype followed by softmax activation. Furthermore, they adapt it to the KD setup by using the teacher logit and feature. They further derive an evidence lower bound to ease the optimization of their reformulation of the classification objective as MOE one. They further pose this MOE based knowledge distillation as an expectation maximization and also discuss its convergence analysis.

Furthermore, to validate their formulation, they perform a nummber of experiments on the benchmark datasets including CIFAR-100, ImageNet-1K etc and compare with existing KD baselines.

**Strengths:**

The issue highlighted in the paper is critical in cases where the teacher is not trained at a very large scale to show good generalization. The reformulation of the problem seems interesting and as mentioned in the paper that although there have been some papers that have utilized these setups jointly, the contribution of this paper by treating students classifier as an MoE layer with the teacher's knowledge is novel and intuitive. Overall method of reformulating classification and then deriving ELBO corresponding to eq. 4 and connecting this to expectation maximization also seems sound and interesting. Experimental analysis also helps in verifying the author claims on the widely popular CIFAR-100 and ImageNet-1K benchmarks which are quite popular in the distillation literature. Multiple architectures have been used as teacher student pairs for analysis along with also showing results on a Linear Probe setup and standard deviations for one of the results have also been provided in the appendix which is also useful to actually understand the quality of results.

**Weaknesses:**

I have some major concerns on the experimental part of this which puts the utility/neccessity of this new formulation to large scale foundation models into question.
1.) In Table 1 and 3 for CIFAR-100 (which is one of the main results) the gains are always less than 1\% except for one case. In the other main result on ImageNet-1K either the gain is less than 1\% or not the best performer only in the top-1Acc. Similar is the case for linear probe experiment (Table 5). Table 6 does have more than 1\% gain in some of the cases but the two datasets are not as large scale and enough to actually conclude that the scheme is effective. In appendix they provide results on CIFAR-100 with std which further increases this concern that can less than 1\% improved accuracy actually advocate for its utility.

2.) Better arhictectures could have been used in the setup as against these older ones. May be like ViTs or pretrained ones like CLIP, LiTs as teachers/students to just understand the effectiveness of this scheme in the lately trending setup. And on similar lines results on more larger scale datasets would be have been pretty useful to understand the implications.

3.) Since no results are provided on the efficiency, we can't even conclude how much useful w.r.t. compute or memory this method is and can solely judge it on the task numbers which are not very convincing. Therefore, evon though the technical contribution is novel I still can't comprehend as to how it can actually be of much use.

**Questions:**

Please refer to the weakness section. Currently, I have voted it marginally below acceptance due taking into account the formulation and results jointly but I would be happy to update my ratings based on the justification in the rebuttal and (partly) based on what other reviewers think. The major primary concern is the weakly convincing experimental section regarding the utility of this approach.

---

### Meta-Review · Area_Chair_h2X3 · 2024-12-20

**Metareview:**

The authors raise an issue in knowledge distillation regarding cases where the ground truth and the teacher’s prediction do not align. To address this, they propose viewing the existing classification objective as a latent MoE (Mixture of Experts) from the teacher's perspective and suggest an optimization method based on the EM (Expectation-Maximization) algorithm. However, the consistency between the problem statement, the proposed algorithm, and the problem-solving approach is lacking. The ablation study is some insufficient.

**Additional Comments On Reviewer Discussion:**

All our reviewers lean to reject this paper. Thanks a lot!

---

### Decision · Program_Chairs · 2025-01-22

Reject